# Flood Resilience of Housing Infrastructure Modeling and Quantification Using a Bayesian Belief Network

**Mrinal Kanti Sen** [1], **Subhrajit Dutta** [1] and **Golam Kabir** [2,*]

1 Department of Civil Engineering, National Institute of Technology Silchar, Assam 788010, India; mksen88@gmail.com (M.K.S.); subhrajit.nits@gmail.com (S.D.)
2 Industrial Systems Engineering, University of Regina, Regina, SK S4S 0A2, Canada
* Correspondence: golam.kabir@uregina.ca

**Abstract:** Resilience is the capability of a system to resist any hazard and revive to a desirable performance. The consequences of such hazards require the development of resilient infrastructure to ensure community safety and sustainability. However, resilience-based housing infrastructure design is a challenging task due to a lack of appropriate post-disaster datasets and the non-availability of resilience models for housing infrastructure. Hence, it is necessary to build a resilience model for housing infrastructure based on a realistic dataset. In this work, a Bayesian belief network (BBN) model was developed for housing infrastructure resilience. The proposed model was tested in a real community in Northeast India and the reliability, recovery, and resilience of housing infrastructure against flood hazards for that community were quantified. The required data for resilience quantification were collected by conducting a field survey and from public reports and documents. Lastly, a sensitivity analysis was performed to observe the critical parameters of the proposed BBN model, which can be used to inform designers, policymakers, and stakeholders in making resilience-based decisions.

**Keywords:** resilience; housing infrastructure; Bayesian belief network; flood hazard and sensitivity analysis

## 1. Introduction

Resilience is defined as the capability of a system to sustain against any hazard and return to its desired performance level after the occurrence of the hazard [1]. Hosseini et al. and Meerow et al. reviewed the definition of resilience in different disciplines [2,3], and its meaning has been discussed and evaluated in the existing literature [3,4]. The reliability and recovery of infrastructure are the two key dependent parameters of infrastructure resilience; furthermore, these two key parameters depend on four additional parameters: robustness, redundancy, rapidity, and resourcefulness, as shown in Figure 1 [1,5–7]. Robustness refers to the sustainability of a system against the effects of the disaster, redundancy refers to the duplication of any critical components or functions of a system that are intended to increase the reliability of the system, rapidity refers to the length of time required to return to its desired position after the occurrence of the hazard, and resourcefulness refers to the availability of resources for recovery. Reliability depends on the robustness and redundancy of the infrastructure, whereas the recovery process depends on rapidity and resourcefulness. Therefore, determining the reliability of infrastructure involves considering parameters based on robustness and redundancy, and similarly, determining the recoverability of infrastructure involves considering parameters based on rapidity and resourcefulness.

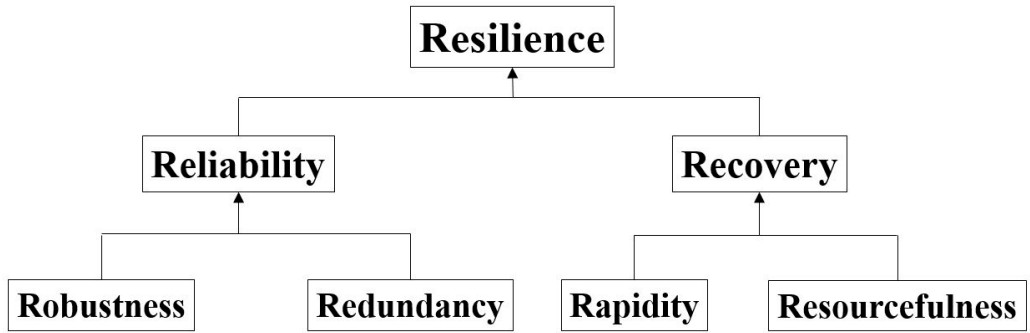

**Figure 1.** Resilience flowchart of the dependency parameters.

Figure 2 represents the generalized performance of a system/infrastructure over its service life [7,8], where A represents the initial condition of a system (which is generally considered to be 100% performance); AB and DE represent the gradual degradation in system performance due to operational conditions; BC represents a sudden drop in system performance due to a disaster, which is also known as loss; $CT_1$ represents the robustness of a system; $T_1T_2$ represents the time required for the recovery of the system; CD represents the recovery profile of the system. Figure 2 shows that, initially, the system/infrastructure performance degrades with time due to natural causes. Then, due to the occurrence of a disaster, the performance level sharply declines. The loss that is shown in the figure mainly depends on the impact of the disaster and the robustness of the system/infrastructure, which means that if the resistance ability of the system is very high, then the losses due to the disaster will be very low. The losses can be estimated using the Hazus technical manual created by the Federal Emergency Management Agency [9]. This manual provides several methodologies for multihazard loss estimation. After the loss, the system tries to recover to its baseline performance by following a recovery profile, which is uncertain and dependent on the type of infrastructure system. There are three types of recovery profiles: linear, non-linear, and stepped. The restoration of roads and bridges, for example, typically follows a stepped recovery pattern.

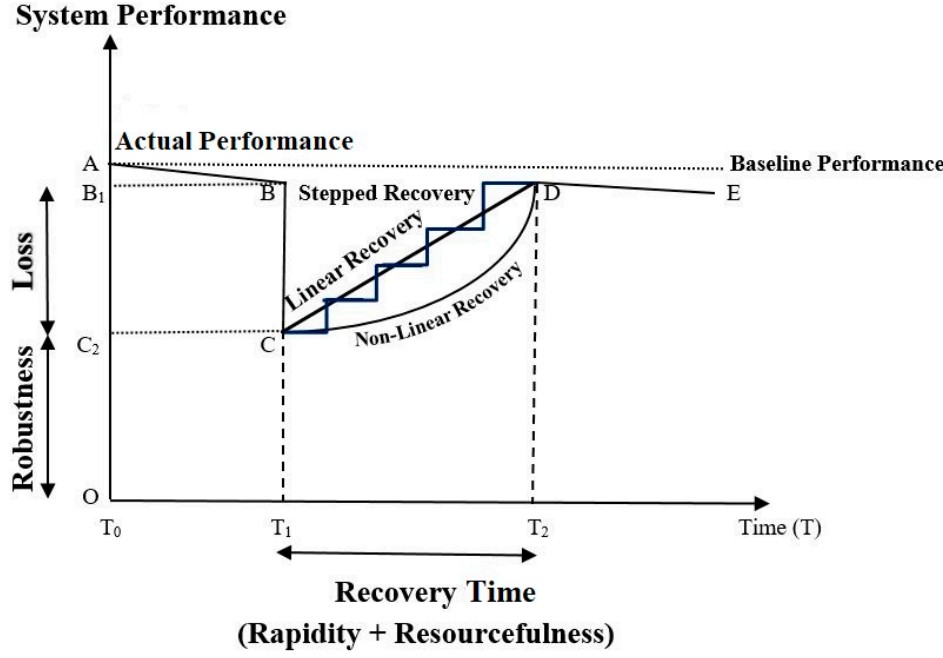

**Figure 2.** System Performance over its service Life.

### 1.1. Socio-Physical Infrastructure

Over the past decade, resilience quantification for communities has been an active area of research for both scientists and engineers. Engineering resilience is relatively new and currently developing, and valuable resources are available for the development of new engineering practices, codes, and regulations [10]. A community is defined as a group of people living in a given geographical area and mainly comprising two key infrastructure systems, namely, social and physical, as shown in Figure 3a [7]. The cross-dependency between different infrastructure systems is shown in Figure 3b [11,12].

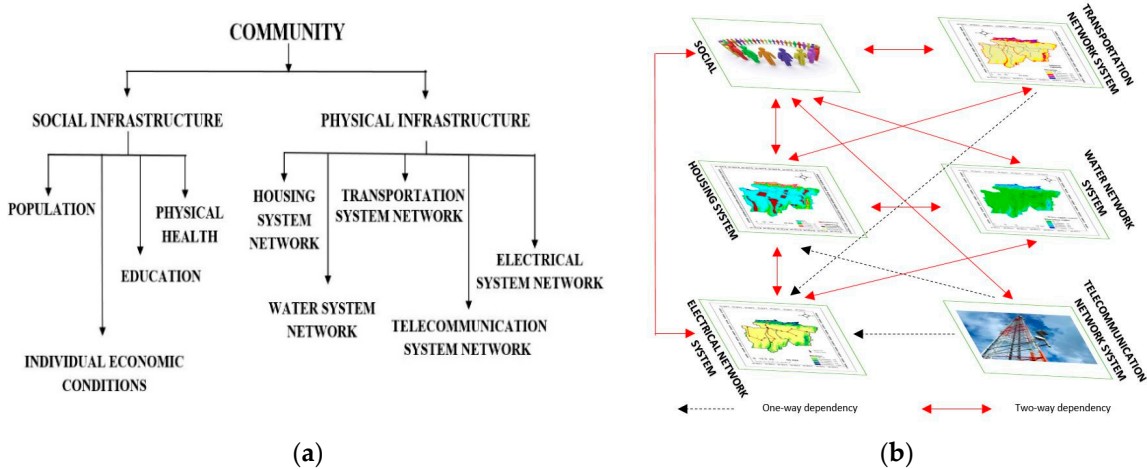

(**a**)  (**b**)

**Figure 3.** Representation of community infrastructure and multilayer interdependency between different infrastructure systems: (**a**) Community infrastructure; (**b**) Interdependency.

Social infrastructure resilience is the ability of societies to resist the effects of a disaster and mainly depends on the population of the community, physical health conditions, literacy or education level, and economic conditions. The physical infrastructure consists of networks for transportation, electricity, water, and telecommunications [7]. The dependencies between infrastructures enhance the complexity of infrastructure resilience quantification, which can be modeled using different approaches [12–14].

Quantification of resilience is very challenging because of many factors, including non-linear relationships between the dependent parameters, a lack of mathematically proven equations or studies to represent these relationships, a requirement for both qualitative and quantitative data, a need for the involvement of experts, data scarcity, data from different sources, and missing data. To overcome these challenges, a resilience measurement scale has been developed for performing quality assessments [15]. Mahmoud and Chulahwat developed a mathematical model to reduce the effect of hazards and proposed a new resilience model for resilience quantification [16]. An infrastructure resilience model plays a crucial role in the proper operation of an infrastructure system during and post disaster in terms of satisfying societal needs [17]. Resilience quantification for water and telecommunication networks was performed using the Resilience-compositional demand/supply (Re-CoDeS) framework [18].

Various frameworks and models have been proposed for quantification and studying resilience in different fields [19–21]. Several methodologies have been proposed for the proper quantification of resilience, such as probabilistic methods [14,22,23], graph theory methods [24,25], fuzzy logic methods [26], and analytical methods [27,28]. A "PEOPLES" (Population and demographics, Environmental and ecosystem, Organized governmental services, Physical infrastructure, Lifestyle and community competence, Economic development, Social cultural capital) factor-based framework for resilience quantification at different scales was also proposed [7,8]. To keep the sustainability of a structure against future hazards, the structure should be resilient enough. Resilience quantification needs a well-formatted past event dataset; however, the biggest hindrance in quantifying resilience

is the lack of availability of properly formatted data for the damage and recovery for infrastructure systems. Inappropriate datasets for previous damage and restoration can lead to inaccurate probability estimations for future disasters and hamper sustainable development. Additionally, dependencies between infrastructure systems play a crucial role in resilience quantification [29]. A virtual system was formed based on an interdisciplinary system to improve resilience and identify the impacts of post-disaster recovery efforts [30]. The codes and standards for designing resilient systems were updated to consider both physical and non-physical infrastructure systems, and a new model for system resilience quantification was developed that considers dependencies and cross-dependencies between the networks, which makes the system more resilient [11]. Resilience has also been discussed and quantified in various networks, such as housing [31], the transportation network [32–34], the electrical network [35,36], the water network [37,38], and the telecommunication network [39,40], but there is a lack of literature that is directly focused on the flood resilience of housing infrastructure systems. Sen et al. studied the resilience of housing infrastructure by using the variable elimination method, but interdependencies between the resilience parameters were not considered in that study, which is a major drawback, as dependency plays a vital role in resilience [31]. This present work is novel in that it directly addressed the housing infrastructure system by considering the dependencies between the resilience parameters against flood hazards.

The main objectives of this work were as follows: (i) to perform a comprehensive study/survey of a community and its socio-physical infrastructure to identify the most influential factors affecting the flood resilience of its housing infrastructure system, (ii) to develop a probabilistic graphical model (Bayesian network model) for the flood resilience quantification of a housing infrastructure system, (iii) to quantify the flood resilience for housing infrastructure against flood hazards, and (iv) to check the sensitivity of each dependent parameter of reliability and recovery.

### 1.2. Socio-Physical Infrastructure of the Barak Valley Community

In this research, the case study region selected was the Barak Valley region of Northeast India. This valley is one of the most important regions of Northeast India as it connects many neighboring states of India. The longitude of this region ranges from 92°15′ E to 93°15′ E, and the latitude ranges from 24°8′ N to 25°8′ N. The total surface area of this valley is approximately 262,230 km$^2$, with a population of more than 3.6 million [41]. The climate of Barak Valley is sub-tropical, warm, and humid, the average rainfall of this valley is 3180 mm, and due to the high intensity of rainfall, floods and landslides are common in the valley. Per the Assam State Disaster Management Authority (ASDMA), in 2017, due to flooding, many water sources were severely damaged, with an estimated restoration cost of more than 277 million USD and an additional 150,000 USD sanctioned for housing system recovery. In 2018, approximately 200,000 people were affected, more than 1300 hectares of agricultural land were damaged, and a main national highway (NH-53, 44) and several other highways remained non-functional for several days. With each year, the damage and costs due to flooding increase [42]. The occurrences of such disasters are frequent in this region, and hence, the associated risk is high [42]. In this valley, nearly 11% of the population do not live in a house, and only 1% of the population live in a house with three or more rooms. The average annual per capita income of the valley is generally low and is in the range of 205 to 342 USD [41,43]. In this study, the housing infrastructure system of Barak Valley was used as the basis for the case study.

Barak Valley is a developing community with mixed demographics and economic conditions. Per the census report, only 30.75% of households use electricity and 0.84% of households use internet services [41]. The elevation of the valley varies from −58 m to 1694 m from mean sea level (MSL), as shown in Figure 4 [44]. In Figure 4, the outlines signify the administrative divisions of an Indian state, which is known as a district; this valley consists of three districts, namely, Cachar, Karimganj, and Hailakandi. Most of the population-dense areas of this valley are located in low-elevation zones, as shown in

Figure 5 [41,44]. Hence, from a flood risk perspective, the valley can be expected to incur significant socio-economic losses.

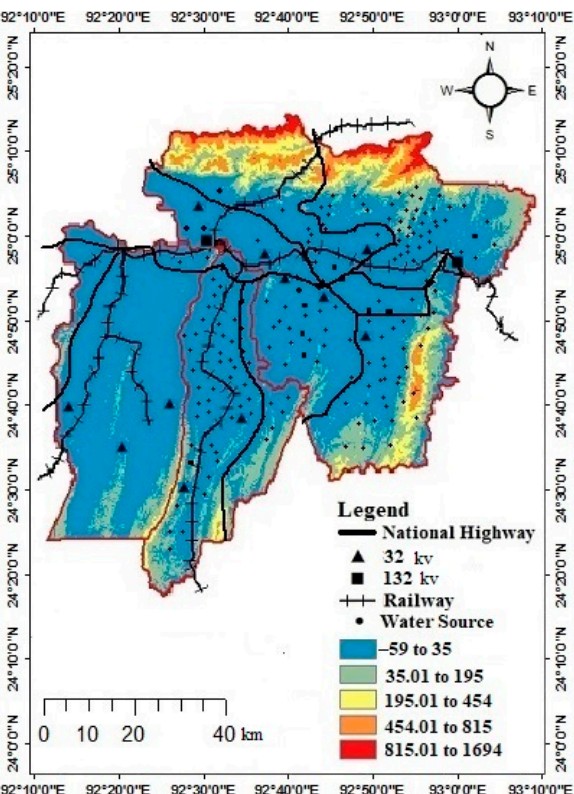

**Figure 4.** Details of Barak Valley overlapping with a digital elevation model (DEM).

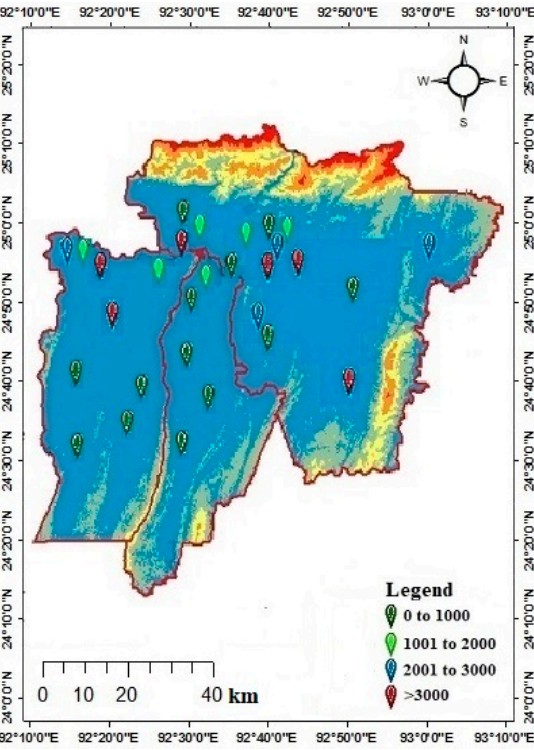

**Figure 5.** Population densities of different areas.

The housing system of this valley comprises various building typologies with a wide range of construction materials. Most of the residential and industrial buildings of this valley are located in low elevations, in the range of −59 to 35 m from MSL, as shown in Figure 6 [44]. These buildings are expected to have a high level of exposure with significant losses during a flood-related disaster. In this valley, traditional single-family houses, also known as Assam-type houses, were found to be the most common type of construction for both urban and rural areas. This type of house is constructed in flat and sloped terrains. The roof is mainly erected using high gables and the walls are made up of timber frames that are plastered with cement and the flooring is made up of either wood or concrete. This type of construction is less reliable and robust compared to RCC construction. More recently, RC construction has increased significantly in urban localities.

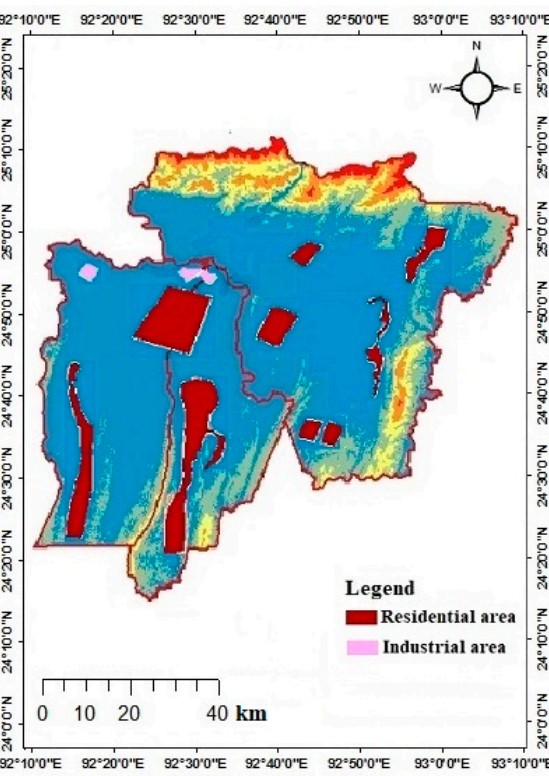

**Figure 6.** Categorization of buildings based on occupancy.

The water system plays a vital role in a community, as its primary function is to provide potable water to residential and commercial buildings. Proper functioning of the overall system depends on the working conditions of individual components, such as the supply source, water pipeline, treatment plant, water tanks, and reservoirs, along with their dependencies. Most of the water supply sources of this valley are located at lower elevations, as shown in Figure 4.

There are three aspects of the electrical power supply network: generation, transmission, and distribution [45]. This network plays a role that is as critical as other infrastructure systems, such as housing, and the water network depends on the electrical network to function. The electrical network comprises five components, which include transmission towers, substations, transformers, electric towers, and electric poles. It can be seen in Figure 7 that the majority of the substations in this valley are located at low elevations, leading to a higher risk of being damaged by floods [46]. On the other hand, the telecommunications network is another important infrastructure system in a community, as the number of phone calls increases during and after any disaster. Most of the population in rural areas do not use internet services, which increases the lack of awareness and communication. For instance, many small communication towers are installed on building roofs, which

may lead to low supply connectivity in an area with a high connectivity demand. Hence, it is expected that the resilience of the communication network and other interdependent systems in Barak Valley will be relatively low during and after disasters.

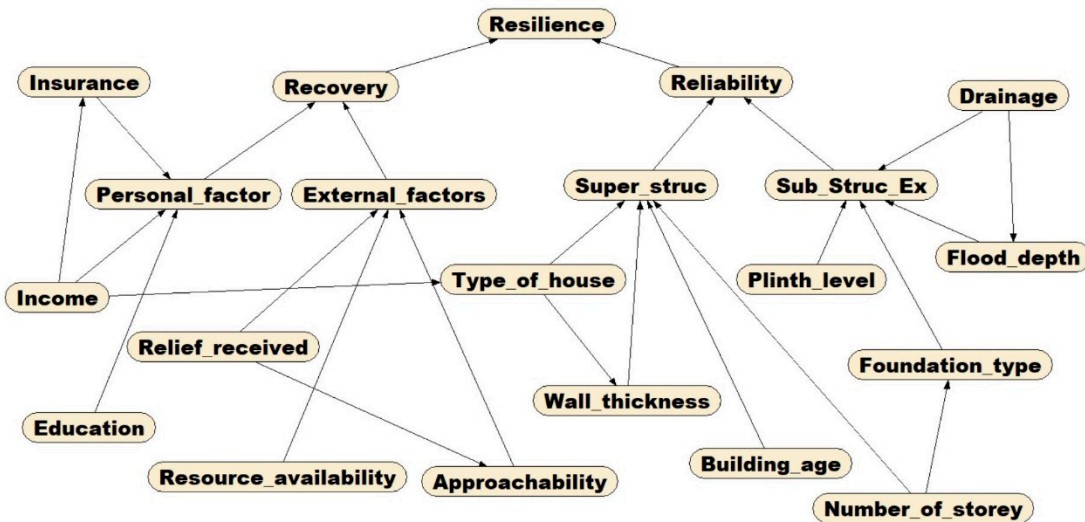

**Figure 7.** Bayesian belief network (BBN) model for flood resilience quantification of a housing infrastructure system.

The transportation network of this valley connects several major states within India. The transportation network comprises four modes: the railway, the roadway, the airway, and the waterway. In the roadway network, bridges are lifelines, as they are considered the most sensitive points of failure during a disaster. Flexible pavement is found to be the most common type of pavement in this valley. Recently, rigid pavement with RC and paver block has become preferred in construction for mitigating repetitive damage due to floods. The airport serving this valley is located at a relatively high elevation, 107 m from the MSL, with a total area of 36.70 acres. Due to the small-scale operation of the airport with limited aircraft, helicopters, and cargo vehicles, it is expected to have a low impact on post-disaster recovery, rescue, and relief operations.

The remaining sections of this paper are organized as follows. In Section 2, the proposed BBN model based on the housing infrastructure system is described in detail. In Section 3, the data collection process for the flood resilience study is described. In Section 4, the proposed model is verified, the sensitivities of the parameters of the proposed BBN model are evaluated, and the proposed BBN model is applied to assess the reliability, recovery, and resilience values of Barak Valley for the housing infrastructure system. Finally, in Section 5, conclusions, limitations of the study, and recommendations for further research are discussed.

## 2. Probabilistic Graphical Model

To develop an effective resilience framework for the housing infrastructure system, it is necessary to utilize different types of data, such as damage and recovery data from multiple sources. Expert judgment should also be obtained for the data interpretation, as the data can often be incomplete. Therefore, it is necessary to consider uncertainties in resilience assessments for the housing infrastructure. To address these uncertainties, different network-based models, such as artificial neural networks (ANNs), an analytic network process (ANP), a Bayesian belief network (BBN), and fuzzy cognitive maps (FCMs) are used. ANN provides insights into uncertainties if a comprehensive post-disaster dataset is available. In the case of insufficient data, techniques such as an ANP, BBN, or FCM can mitigate such uncertainties. It becomes very difficult for experts to generate a supermatrix, as found in the ANP method, where the representation of the relationship between parameters is performed using pairwise comparisons [47]. FCMs allow for the

expression of dependence between the nodes with an influence degree ranging from "+1" to "−1" [48]. BBN assigns effective relationships between the nodes by considering a conditional probability table (CPT). A comparison of these different techniques is shown in Table 1 [4]. Note that, in the table, VH—very high, H—high, M—medium, and L—low. Based on this comparison, the BBN tool was selected for this study.

**Table 1.** Comparison between different techniques.

| Attributes | ANN | ANP | BBN | FCM |
|---|---|---|---|---|
| Capability to express causality | N | L | VH | H |
| Capability to control qualitative inputs | N | VH | H | VH |
| Capability to control quantitative inputs | VH | L | M | L |
| Capability to control dynamic data | H | M | H | M |
| Capability to model complex systems | VH | M | VH | H |
| Learning/training capability | VH | H | H | H |

ANN: artificial neural network, ANP: analytic network process, BBN: Bayesian belief network, FCM: fuzzy cognitive map; L: low, M: medium, H: high, VH: very high, N: Negligible.

## 2.1. Bayesian Belief Network

BBN is an extensive probabilistic model that is used to characterize the uncertainty that is associated with variables that constitute the model [49]. BBN is a graph-based model comprising nodes and edges, where nodes represent model variables and edges that represent the relationship between the nodes and also the conditional dependencies [50,51]. BBN can also be defined as a class of graphical models that presents a brief representation of the probabilistic dependencies between a given set of random variables [52]. The BBN model generally deals with discrete probabilities; therefore, each node is categorized into a finite set of variables with their probability values [53]. The CPT quantifies the dependencies between the child node and the parent node, as shown in Figure 8, where node A represents the child node and nodes B and C represent the parent nodes. The CPT for the parent nodes transfers to the unconditional probability (UP) and can be attained via an expert decision [53] and/or training from the dataset [54].

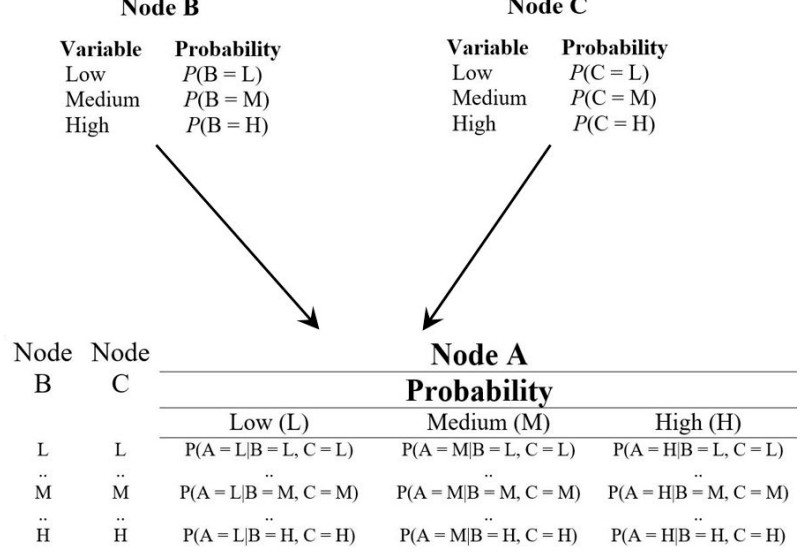

**Figure 8.** Representation of a BBN.

Probabilistic methods, such as a BBN, have been used to model various types of networks, such as transportation networks [32,55–57], water networks [14,38], telecommunications networks [58,59], and electrical networks [60]. An infrastructure system can be depicted as a BBN by using the dependent parameters and their interdependencies. BBN

modes have also been used in the assessment of risks in infrastructure systems [61,62]. In this model, variables of any parameter can be updated when the values of evidence variables are available [63]. BBN comprises two types of inference, such as forward inference and backward inference, where the forward inference method is used to determine the system's probability value based on its component probability values and the backward inference method is used to determine the value of the updated probability of a system or its components based on the system state. It can also simulate the dependence of a network of uncertain variables and compute the inferences of any event based on evidence. For example, consider a BBN in which the probability occurrence of an event $A$ depends on the occurrences of any other event, such as $E_i$ ($i = 1, \ldots, n$). The posterior probability of the $i$th event, $E_i$, given the occurrence of $A$, is given by Bayes' theorem, as shown in Equation (1) [64]:

$$P(E_i|A) = \frac{P(A|E_i)P(E_i)}{P(A)},$$ (1)

where $P(A \mid E_i)$ is the conditional probability of the event $A$ given the occurrence of $E_i$ and $P(E_i)$ is the prior probability of $E_i$.

### 2.2. BBN Model for Flood Resilience Quantification of Housing Infrastructure

In this section, the development of the BBN model for flood resilience is discussed. Initially, several experts from various domains were selected. Then, based on the experts' knowledge and the literature, the resilience parameters were selected. The experts were selected based on the assumption that they have experience in the field of utility infrastructure management, risk management, Bayesian analysis, or catastrophes. A total of ten experts were selected for this work; their experience was as follows: (i) two field officers from the District Disaster Management Authority (DDMA) with more than three years of experience and who also helped during the field survey for data collection; (ii) one district project officer (DPO) from DDMA with more than ten years of experience and expertise in disaster management monitoring and policy implementation; (iii) two academic experts from different institutions having more than two and ten years of experience, respectively, with both obtaining a Ph.D. in Civil Engineering and are experts in reliability, risk, and resilience assessments; (iv) five flood catastrophe modelers from different industries with experiences of 16, 7, 5, 5, and, 4 years, respectively, where all of whom obtained a Ph.D. in Civil Engineering. Several meetings were conducted with the experts for the selection of resilience parameters and the identification of interdependencies. As resilience depends on the two key factors of reliability and recovery, 14 resilience parameters for housing infrastructure against flood hazards were selected based on the experts' knowledge and a literature review. The selected resilience parameters were as follows: (i) Type_of_house (robust types of houses are expected to perform better during floods) [31,65]; (ii) Wall_thickness (increasing wall thickness increases the resistance capability and reliability) [31,66]; (iii) Building_age (newly constructed buildings were found to be in better condition than older building) [31,67]; (iv) Number_of_stories (as stories increase, casualties decrease during floods) [31,67]; (v) Drainage (adequate drainage systems prevent damage to infrastructure) [31,68]; (vi) Flood_depth (this produces water pressure, which reduces the reliability of the infrastructure) [31,68]; (vii) Foundation_type [65,69]; (viii) Plinth_level (increasing the plinth height of the house to the top level of the road reduces vulnerability) [31]; (ix) Insurance (insurance enhances the recovery rate as the insurer can pay off the insurance claims) [31,70]; (x) Income (for households with higher income levels, the recovery process will be faster after a disaster) [31,69,71]; (xi) Education (education enhances the preparedness against disaster) [67,69,72]; (xii) Relief_received (whether during/after the disaster relief is received or not) [31,73,74]; (xiii) Approachability (whether connectivity from resource location to vulnerable site is disturbed or not) [31,69]; (xiv) Resource_availability (whether during/after the disaster raw materials for construction are available locally or not) [31].

To reduce the complexity during the construction of the CPT, reliability was subdivided into two parameters: (i) superstructure condition (Super_Struc) and (ii) substructure and external conditions (Sub_Struc_Ex). Next, these two parameters were linked with different resilience parameters, such that Super_Struc was linked with Type_of_house, Wall_thickness, Building_age, and Number_of_stories, while Sub_Struc_Ex was linked with Drainage, Flood_depth, Foundation_type, and Plinth_level. Similarly, recovery was divided into two parameters: (i) Personal_factor and External_factor. Personal_factor was then linked with Insurance, Income, and Education, while External_factor was linked with Relief_received, Approachability, and Resource_availability. Finally, the two key parameters, namely, reliability and recovery, were linked with resilience, as dependency plays a crucial role in resilience quantification. Therefore, based on the experts' judgements, the dependency was constructed and the dependencies between the parameters were as follows: (i) Type_of_house depended on Income, (ii) Wall_thickness depended on Type_of_house, (iii) Flood_depth depended on Drainage, (iv) Relief_received depended on Approachability, (v) Insurance depended on Income, and (vi) Foundation_type depended on Number_of_stories. After the selection of resilience parameters and the assignment of interdependencies, a BBN model was developed.

The developed BBN model for the flood resilience quantification of housing infrastructure with dependencies is shown in Figure 7. In the model, resilience was categorized into three different probability states—low (L), medium (M), and high (H)—where low means that immediate attention should be given by the stakeholders to the housing infrastructure of the community for strengthening or reconstruction, medium means that the housing infrastructure of the community can act as functional for the long-term, and high means safety exists in the housing infrastructure of the community for future hazards [75]. Reliability was categorized into four different probability states: DS1, DS2, DS3, and DS4, as discussed in Table 2 [76].

**Table 2.** Damage state (DS) descriptions.

| Damage State | Category | Description |
|---|---|---|
| DS1 | Low | No-damage condition, where floodwater touches the foundation but has no contact with electrical systems with a water height that is about 2.5 cm from ground level and damage occurs to carpets and flooring. |
| DS2 | Medium | Drywall damage up to a 30 cm water level from the ground level and damage occurs to household furniture and other major equipment on the floor; doors need to be replaced. |
| DS3 | High | Electrical panels, bathroom/kitchen cabinets and electrical appliances, lighting fixtures on walls, ceiling lighting, and studs got damaged. |
| DS4 | Very High | The structure is fully damaged. |

DS4 means that damage due to the occurrence of the flood is very high, which indicates that the reliability of the housing infrastructure of the community is very low; on the other end of the range, DS1 means damage due to the occurrence of the flood is very low, which indicates that the reliability of the housing infrastructure of the community is very high. Recovery is categorized into three different probability states—low (L), medium (M), and high (H)—where H is less than 10 days, which corresponds to the 25th percentile for recovery time; M is between 11 and 35 days, which corresponds to the 26th to 75th percentile for recovery time; L is more than 35 days, which corresponds to the 76th to 100th percentile for recovery time. The recovery of the housing infrastructure is discussed based on the amount of time required for the infrastructure to fully recover, where high means that the infrastructure took little time to fully recover and low means that the infrastructure took a very long time to fully recover. The recovery time is mainly based on the recovery

of the infrastructure; it does not include the recovery of housing essentials, such as kitchen essentials, bathroom essentials, furniture, and utilities. The different probability states of each parent and intermediate parameter, as shown in Tables 3 and 4, were assigned based on the field survey data and experts' knowledge.

**Table 3.** Variable details for the parent parameters.

| Parameter | Scale | Parameter | Scale |
|---|---|---|---|
| Annual Income (Indian Rupees) | Less than 10,000, 10,000–20,000, or more than 20,000 | Education | Below 10th standard, 10th or 12th standard passed, or graduate |
| Insurance | Yes or no | Relief_received | Yes or no |
| Resource_availability | No, yes with a 0 to 10% increase, or yes with more than a 10% increase | Approachability | Yes or no |
| Building_age | Less than 10 years, 10 to 20 years, or more than 20 years | Plinth_level of house w.r.t the road top level | Up to 1 m, 1 to 2 m, and above 2 m |
| Drainage availability | Yes or no | Number_of_stories | 1 or more than 1 |

**Table 4.** Variable details for the intermediate parameters.

| Parameter | Scale | Parameter | Scale |
|---|---|---|---|
| Personal_factor | Good or bad | Super_Struc | Good or bad |
| External_factor | Good or bad | Sub_Struc_Ex | Good or bad |
| Foundation_type | Shallow or deep | Flood_depth | Less than 30 cm, 30 to 90 cm, or more than 90 cm |
| Type_of_house | Bamboo, masonry (Assam type), or Reinforced Cement Concrete (RCC) | Wall_thickness | Less than 5 cm, 5 to 10 cm, or more than 10 cm |

## 3. Data Collection for Flood Resilience Study

In Barak Valley, flood hazards occur at regular intervals, which mainly affect the infrastructure systems and a considerable rise in annual rainfall in the past few years has contributed to flooding. It has already been stated that for accurate resilience quantification, a properly formatted dataset of past disasters should be utilized; however, the collection of such data is often a difficult task.

As India is a developing country, the proper collection and management of pre- and post-disaster data for housing should be readily available for most governmental agencies. Post-disaster data has been provided by some governmental agencies, but the volume of data provided is inadequate as their variables differ from those in this work. For example, the datasets divided the housing infrastructure into two types, namely, pucca and kutcha, and the number of damaged houses in terms of pucca and the kutcha type was provided. In our work, however, we required the data in terms of bamboo, masonry, and RC type. Moreover, information on the maximum considered nodes in this BBN model is unavailable.

To overcome this challenge and to acquire the necessary data, an extensive field survey was performed in various flood-affected areas in this valley regarding housing infrastructure systems. A flood resilience assessment form was prepared for the survey, as shown in Figure 9. In this process, we visited 23 vulnerable places and 1 non-vulnerable

place. Post-disaster data were collected for more than 500 houses, with each survey taking around 20–25 min.

**Figure 9.** Flood resilience assessment form.

During the survey, respondents provided recommendations for future preparedness, which we discuss in the last section of this document. The form was designed in a generalized manner, such that it can be used for resilience quantification of housing infrastructure systems in different communities. This collected data and associated data-driven resilience analysis are beneficial for improving the preparedness of a housing infrastructure system for future flood disasters, increasing its structural reliability, and enabling a thorough risk assessment against flood hazards.

## 4. Results and Discussion

### 4.1. Model Validation

The validation of a model is critical, as an inaccurate model will always provide erroneous results. Therefore, it is very important to validate the proposed BBN model such that it can provide accurate information. In this study, two qualitative validation approaches, namely, an extreme condition test and a scenario analysis, were performed to validate the proposed BBN model [77]. The experts' judgment played a critical role in

developing and validating the proposed BBN model due to data scarcity. Initially, the prior probability of each parent node was assigned from the collected field survey dataset. As an example, consider $P(Y)$, the prior probability of "$Y$" and, say, there are two variables for $Y$, namely, "yes" and "no." If $X$ is the total number of data collected, out of which, "$A$" is the number of data for the "no" variable and "$B$" is the number of data for the "yes" variable, then the prior probability of "$Y$" is assigned based on Equation (2) [78]:

$$P(Y = \text{no}) = A/X \text{ and } P(Y = \text{yes}) = B/X, \qquad (2)$$

and the CPT between the parameters is obtained based on expert judgment and the collected data. During the development of the CPT, the DPO of DDMA and other industry experts generated the CPT values for each intermediate and child node based on their knowledge and the available literature. Next, the academic experts modified the CPT values according to their experience, and finally, CPT values were assigned between the parent and child node.

In this study, 332 CPT values were generated for the proposed BBN model. The recovery, reliability, and resilience values for different vulnerable places were computed using Netica software [79].

### 4.1.1. Extreme Condition Analysis

In the extreme condition analysis, two extreme conditions, namely, extreme 1 (E-1) and extreme 2 (E-2), were considered for the analysis in this process. E-1 represented one of the most vulnerable places (Burunga) in this valley, where all the parent nodes of resilience were in the worst condition states, while E-2 represented a non-vulnerable place (Tarapur), where all the parent nodes were in favorable condition states. The proposed BBN model for resilience based on the housing infrastructure system was applied to these two extreme conditions. Here, for the E-1 and E-2 conditions, the recovery for Burunga and Tarapur was estimated as being (low, medium, high) = (75.9, 15.7, 8.38) and (24.5, 26.6, 48.9), respectively. Similarly, the reliability for Burunga and Tarapur was estimated as (DS1, DS2, DS3, DS4) = (12.2, 15.2, 21.6, 51.0) and (33.2, 24.6, 22.1, 20.1), respectively. The resilience for Burunga and Tarapur was estimated as (low, medium, high) = (66.9, 18.4, 14.7) and (29.3, 25.7, 45.0), respectively. The evaluated values indicated that E-1 and E-2 had the highest probabilities of 66.9% and 45.0% at the low and high resiliencies, respectively. These indicated that for the E-1 condition, the probability of resilience in the low state was higher than other states, but for the E-2 condition, the probability of resilience was in the high state was higher. The E-1 and E-2 tests showed that the proposed BBN model based on the housing infrastructure system performed according to the assumed model behavior, which also indicated that the proposed BBN model was valid.

### 4.1.2. Scenario Analysis

In the scenario analysis, five different types of scenarios were considered for the quantification of reliability and recovery. The probability states of all resilience parameters for all five scenarios are presented in Tables 5 and 6. Scenario 1 represents all the parent nodes in the severe condition; with subsequent progress of the parent nodes, the conditions improved gradually to ultimately reach the best condition, as represented by scenario 5. The probability states for the resilience parameters for each scenario were assigned based on the condition of the scenario. Based on the probability states, the reliability and recovery for all the scenarios were calculated, as shown in Tables 5 and 6.

**Table 5.** Scenario analysis for recovery.

| Parameter | Scenario 1 | Scenario 2 | Scenario 3 | Scenario 4 | Scenario 5 |
|---|---|---|---|---|---|
| Insurance | No | No | No | No | Yes |
| Income | Less than 10,000 | 10,000 to 20,000 | 10,000 to 20,000 | More than 20,000 | More than 20,000 |
| Education | Below 10th standard | 10th or 12th standard passed | 10th or 12th standard passed | Graduate | Graduate |
| Relief_received | No | No | Yes | Yes | Yes |
| Resource_availability | No | Yes with more than a 10% increase | Yes with more than a 10% increase | Yes with more than a 10% increase | Yes with an increase by 0 to 10% |
| Approachability | Yes | Yes | No | No | No |
| **Recovery** | | | | | |
| Low | 74.3 | 50.9 | 32.9 | 26.5 | 1.7 |
| Medium | 18.3 | 29 | 35.1 | 33 | 11.9 |
| High | 7.4 | 20.1 | 31.9 | 40.5 | 86.4 |

**Table 6.** Scenario analysis for reliability.

| Parameter | Scenario 1 | Scenario 2 | Scenario 3 | Scenario 4 | Scenario 5 |
|---|---|---|---|---|---|
| Type_of_house | Bamboo | Assam Type | RCC | RCC | RCC |
| Wall_thickness | Less than 5 cm | 5 to 10 cm | More than 10 cm | More than 10 cm | More than 10 cm |
| Building_age | More than 20 | More than 20 | Up to 20 | Up to 10 | Up to 10 |
| Number_of_floor | 1 | 1 | More than 1 | More than 1 | More than 1 |
| Plinth level | More than 2 m | More than 2 m | More than 2 m | Up to 1 m | Up to 1 m |
| Foundation_type | Shallow | Shallow | Shallow | Deep | Deep |
| Flood_depth | More than 90 cm | More than 90 cm | More than 90 cm | 30 to 90 cm | Less than 30 cm |
| Drainage | No | No | No | No | Yes |
| **Reliability** | | | | | |
| DS1 | 4.5 | 9.9 | 17.5 | 28.3 | 87.2 |
| DS2 | 7.8 | 16.1 | 27.8 | 28.7 | 7.8 |
| DS3 | 30.9 | 29.4 | 27.3 | 23 | 3.3 |
| DS4 | 56.8 | 44.6 | 27.4 | 20 | 1.7 |

In scenario 1, the probabilities of recovery and reliability were (low, medium, high) = (74.3, 18.3, 7.4) and (DS1, DS2, DS3, DS4) = (4.5, 7.8, 30.9, 56.8); in scenario 2, the probabilities of a low state recovery and DS4 state reliability decreased from 74.3 to 50.9 and from 56.8 to 44.6, respectively, while the probabilities of a high state recovery and DS1 state reliability increased from 7.4 to 20.1 and from 4.5 to 9.9, respectively; in scenario 3, the probabilities of a low state recovery and DS4 state reliability decreased to 32.9 and 27.4, respectively, while the probabilities of a high state recovery and DS1 state reliability increased to 31.9 and 17.5, respectively; in scenario 4, the probabilities of a low state recovery and DS4 state reliability decreased to 26.4 and 20.0, respectively, while the probabilities of a high state recovery and DS1 state reliability increased to 40.5 and 28.3, respectively; in scenario 5, the probabilities of a low state recovery and DS4 state reliability decreased to 1.7 and 1.7, respectively, while the probabilities of a high state recovery and DS1 state reliability increased to 86.4 and 87.2, respectively. All five scenarios represented the desired model behavior. Similarly, different combinations of parameters were considered to generate different scenarios and

their recovery and reliability probability distributions were tested to perform the model validation. Based on the results of the analysis and the discussion presented above, it is believed that the proposed BBN model was validated.

The validation of the proposed BBN model using different approaches also indicated that the constructed CPTs based on the experts' knowledge were correct. Hence, this model can be used for resilience quantification for housing infrastructure against flood hazards.

### 4.2. Sensitivity Analysis

Sensitivity analysis in the BBN is broadly concerned with understanding the relationship between local network parameters and global conclusions drawn from the network [80–84]. A sensitivity analysis was performed to achieve the following objectives: (i) to identify the critical parameters for reliability and recovery of a housing infrastructure system and (ii) to identify the possible changes of the dependent parameters in the BBN model that can ensure the satisfaction of a query constraint for the target reliability, recovery, or resilience. Sensitivity analysis provides essential information about the results and their variance according to a very small change in the input value with uncertainty [53,80]. This analysis included an investigation of the effect of changes in uncertain input parameters on the uncertainty of the response of interest. The sensitivity analysis also reduced the predicted uncertainty as it identified the high-impact parameters. Here, the variance reduction (VR) method was utilized to identify the sensitivity of the parameters of the proposed BBN model based on the housing infrastructure system [85,86]. This method computes the VR of the expected real value of a query node $R$, for example, reliability and recovery, due to a result that was caused by changing variable node $P$, such as Drainage, Type_of_house, Income, or Resource_availability. Therefore, the variance of the real value of $R$ given evidence on $P$, namely, $V(R|q)$, can be computed using Equation (3) [84]:

$$V(R|q) = \sum_z p(r|q)[Y_r - E(R|q)]^2, \tag{3}$$

where $r$ is the state of the query node $R$, $q$ is the state of the varying variable node $P$, $p(r|q)$ is the conditional probability of $r$ given $q$, $Y_r$ is the value corresponding to state $r$, and $E(R|q)$ is the expected real value of $R$, after the new finding $q$ for node $P$.

The VR and percentage of VR of the parent nodes for the child node recovery and reliability are shown in Figure 10. For the recovery, Insurance showed the highest contribution (2.87%) to the percentage of VR, followed by Relief_received (2.13%), Income (1.05%), Approachability (0.91%), Resource_availability (0.79%), and Education (0.08%). It can be observed that the parameters Education and Resource_availability were far less sensitive for recovery. Similarly, regarding reliability, Type_of_house showed the highest contribution (5.06%) to the percentage of VR, followed by Wall_thickness (4.69%), Drainage (1.53%), Flood_depth (1.04%), Building_age (0.63%), Number_of_stories (0.29%), Foundation_type (0.26%), and Plinth_level (0.19%). It can be observed that the parameters Plinth_level, Foundation_type, and Number_of_stories were far less sensitive regarding reliability.

The sensitivity analysis aligned with the expert statements, as recovery was highly dependent on External_factor (24.1%) and Personal_factor (15%), followed by Insurance and Relief_received, as it is known that the recovery for insured houses is relatively fast; similarly, after a disaster, if a stakeholder provides relief to vulnerable places, then the recovery process can be fast. Reliability was highly dependent on Super_Struc (37.3%) and Sub_Struc_Ex (28.4%), followed by Type_of_house and Wall_thickness, as the resisting ability of RC houses (Type_of_house) against flood hazards is greater compared to bamboo houses, and with an increase of wall thickness, the withstanding capability against hazards increases. Lower wall thickness impacts the Super_Struc, which directly impacts the reliability of the housing infrastructure system, and finally, affects the resilience of the system. It can be observed from the outcome of this analysis that the sensitivity of the child node highly depended on the variability of the parent nodes. This technique also provided information for optimal changes to parameters that were required to obtain

a targeted recovery, reliability, and resilience of an infrastructure system. The crucial parameters to recovery and reliability were identified, thereby providing information for tuning those sensitive parameters for increasing the reliability of systems and to speed up the recovery process.

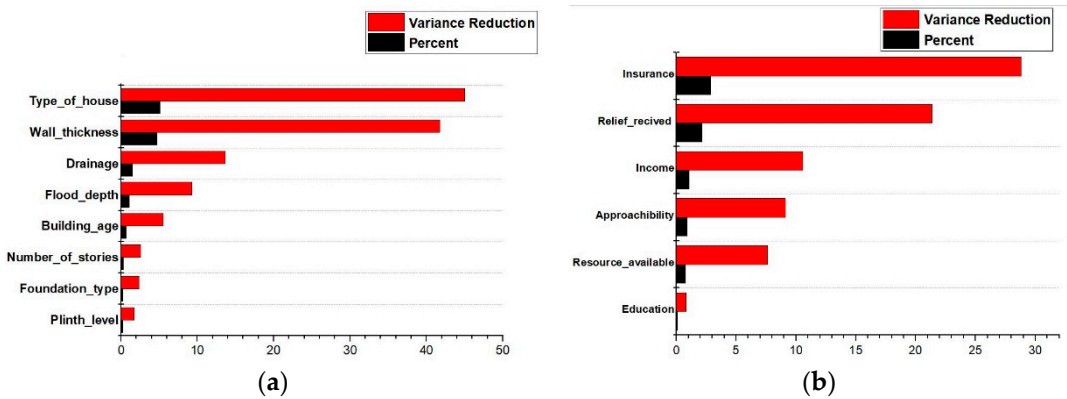

**Figure 10.** Sensitivity analysis for (**a**) reliability and (**b**) recovery.

### 4.3. Reliability, Recovery, and Resilience Values

In this section, the evaluated reliability, recovery, and resilience of the housing infrastructure for Barak Valley is discussed. In this work, the evaluated resilience values considering different parameters represent the resistance to flood hazards and recoverability after the occurrence of the hazard. Tables 7 and 8 show the reliability, recovery, and resilience of Barak Valley. It can be observed from Table 7 that the reliability of more than 50% of the housing infrastructure of Burunga (one of the locations visited) fell under the DS4 state, which indicates that the reliability of that location was extremely low. Similarly, locations such as Poschim Kumrapara, Algapur, and Rajnagar fell under the same probability state. The housing infrastructure recovery of Burunga was also not positive, as can be observed from Table 8, as more than 75% of the housing infrastructure recovery fell under the low probability state. It has been stated that resilience is a combination of recovery and reliability.

It can be observed from Table 8 that the housing infrastructure resilience of Burunga had a maximum probability in the low state. It can be noted, by including the dependencies between the resilience parameters in the model, the results changed compared to the evaluated results by Sen et al. [31]. According to Sen et al., the probabilities of housing infrastructure (Algapur) were 0, 0.1, 0.6, and 0.3, but in this study, the evaluated values were 12.7, 15.3, 21.8, and 50.1 [31]. During the field visit, it was observed that the maximum of the housing infrastructure was in an extremely hazardous situation, which means that the evaluated values of this study were more reliable than the earlier study. Similarly, for Tarapur, it was observed that there were some houses that needed immediate attention in terms of strengthening, but in the earlier study, the probability of housing infrastructure in the DS4 state was zero. In this study, 20.1% of the housing infrastructure of this place was in the DS4 state, which indicated that this study provided more accurate results. Overall, this indicates that the BBN approach was better than the VE method.

**Table 7.** Reliability values of all flood vulnerable areas.

| Place Name | DS1 | DS2 | DS3 | DS4 |
|---|---|---|---|---|
| Algapur | 12.7 | 15.3 | 21.8 | 50.1 |
| Amjurghat | 13.6 | 16.1 | 21.8 | 48.5 |
| Anipur Grant | 16.4 | 18 | 21.9 | 43.7 |
| Baleswar | 16.9 | 18.2 | 21.9 | 43 |
| Bhatirkupa | 19.2 | 18.9 | 22 | 39.8 |
| Borbond | 17.5 | 18.3 | 22 | 42.2 |
| Burunga | 12.2 | 15.2 | 21.6 | 51 |
| Dullabcherra | 21.1 | 20.3 | 21.7 | 36.9 |
| Dwarbond | 19.8 | 19 | 25 | 36.1 |
| Fanai Cherra Grant | 16.9 | 19 | 21.9 | 42.9 |
| Hailakandi Town | 20.7 | 21 | 21.7 | 36.6 |
| Jamira | 17.2 | 18 | 22 | 42.8 |
| Kanakpur | 18.8 | 18.6 | 22.1 | 40.5 |
| Katlicherra | 16.6 | 18.5 | 21.8 | 43.1 |
| Lalaghat | 11.3 | 14.5 | 21.5 | 52.8 |
| Rajnagar | 12.7 | 15.6 | 21.7 | 50.1 |
| Panchgram | 19.8 | 19.7 | 21.8 | 38.7 |
| Poschim Kumarpara | 11.1 | 15 | 21.2 | 52.8 |
| Rakhal Khalerpaar | 13.5 | 17 | 21.6 | 47.6 |
| Rangirghat | 17.5 | 18.7 | 21.8 | 41.9 |
| Ratnapur | 16.3 | 18 | 21.9 | 43.8 |
| Silchar Municipality | 15.7 | 17.4 | 21.9 | 45 |
| Tarapur | 33.2 | 24.6 | 22.1 | 20.1 |
| Uttar Krishnapur | 18.2 | 18.9 | 21.9 | 41 |

**Table 8.** Recovery and resilience values of all flood-vulnerable areas.

| Place Name | Recovery | | | Resilience | | |
|---|---|---|---|---|---|---|
| | Low | Medium | High | Low | Medium | High |
| Algapur | 54.2 | 25.5 | 20.2 | 54.9 | 23.2 | 21.9 |
| Amjurghat | 57 | 24.3 | 18.7 | 55.9 | 22.7 | 21.4 |
| Anipur Grant | 52.1 | 22.4 | 21.9 | 51.5 | 23.8 | 24.7 |
| Baleswar | 53.8 | 25.1 | 21.1 | 52.2 | 23.6 | 24.2 |
| Bhatirkupa | 60.8 | 23 | 16.2 | 54.7 | 23 | 22.2 |
| Borbond | 59.8 | 23.8 | 16.4 | 55.1 | 23.1 | 21.8 |
| Burunga | 75.9 | 15.7 | 8.38 | 66.9 | 18.4 | 14.7 |
| Dullabcherra | 54.8 | 24.3 | 20.9 | 50.4 | 24 | 25.6 |
| Dwarbond | 56.4 | 24.1 | 19.5 | 51.6 | 23.8 | 24.6 |
| Fanai Cherra Grant | 56.2 | 23.9 | 19.9 | 53.3 | 23.2 | 23.4 |
| Hailakandi Town | 46.8 | 23.6 | 26.9 | 46.1 | 25 | 28.9 |
| Jamira | 46.6 | 26.3 | 27.1 | 48.1 | 24.5 | 27.4 |
| Kanakpur | 60 | 23.4 | 16.6 | 54.6 | 23.1 | 22.4 |
| Katlicherra | 53.3 | 24.7 | 22 | 51.9 | 23.6 | 24.5 |
| Lalaghat | 56.2 | 24.9 | 18.9 | 56.9 | 22.5 | 20.5 |
| Rajnagar | 60.1 | 22.7 | 17.2 | 58.1 | 21.8 | 20.1 |
| Panchgram | 66.3 | 20.3 | 13.4 | 57.1 | 22 | 20.9 |
| Poschim Kumarpara | 68.1 | 19.5 | 12.4 | 63.5 | 19.8 | 16.7 |
| Rakhal Khalerpaar | 54.8 | 24.3 | 20.9 | 54.4 | 22.9 | 22.7 |
| Rangirghat | 48.8 | 25.8 | 25.3 | 59 | 24.3 | 26.7 |
| Ratnapur | 49.1 | 25.8 | 25.1 | 49.8 | 24.1 | 26.1 |
| Silchar Municipality | 53.6 | 24.9 | 21.4 | 52.8 | 23.5 | 23.8 |
| Tarapur | 24.5 | 26.6 | 48.9 | 29.3 | 25.7 | 45 |
| Uttar Krishnapur | 59.2 | 23 | 17.9 | 54.2 | 23 | 22.8 |

As most of the surveyed areas are vulnerable to flood hazards, it is clear that the recovery, reliability, and resilience values for the housing infrastructure system were

categorized as low, DS4, and low, respectively. To obtain more detailed information for the reliability, recovery, and resilience of the housing infrastructure of Barak Valley, the "low" state of recovery and resilience, and the DS4 state of reliability were further sub-divided into four additional categories according to the percentile of the total low state values (higher to lower): extremely low (≥75th percentiles), very low (50th to 74th percentile), moderate–low (26th to 49th percentile), and low (≤25th percentile). Finally, three types of flood models—a flood recovery model, flood reliability model, and flood resilience model—of the valley were prepared based on the categorization of the low state, as shown in Figures 11–13.

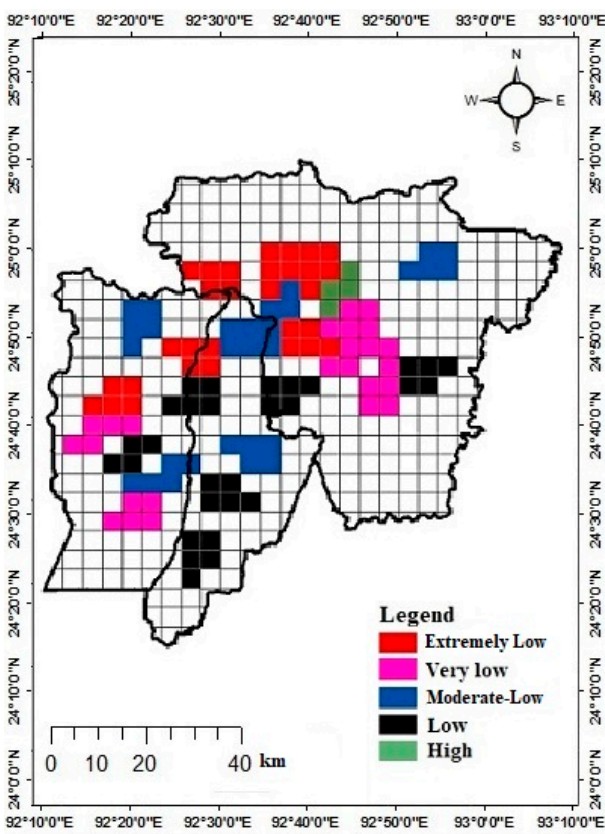

**Figure 11.** The flood recovery model of the housing infrastructure in Barak Valley.

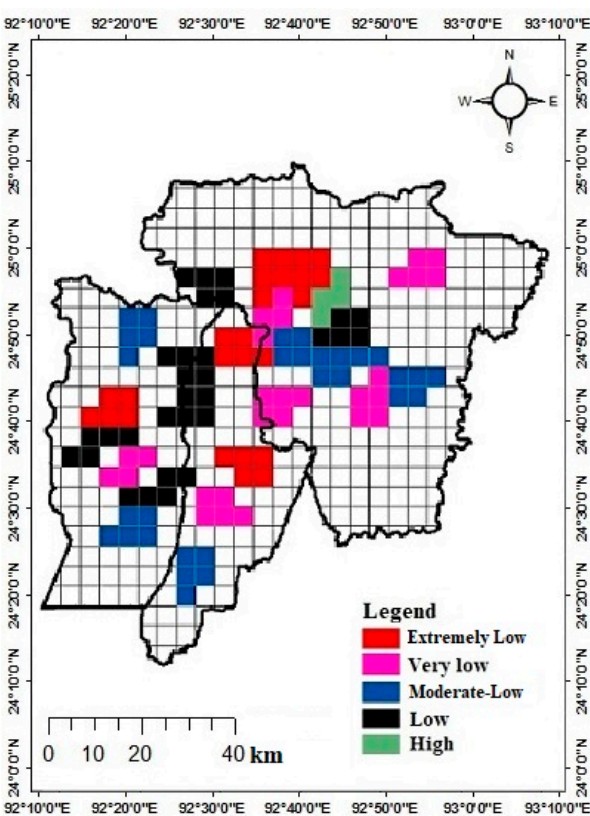

**Figure 12.** The flood reliability map of the housing infrastructure in Barak Valley.

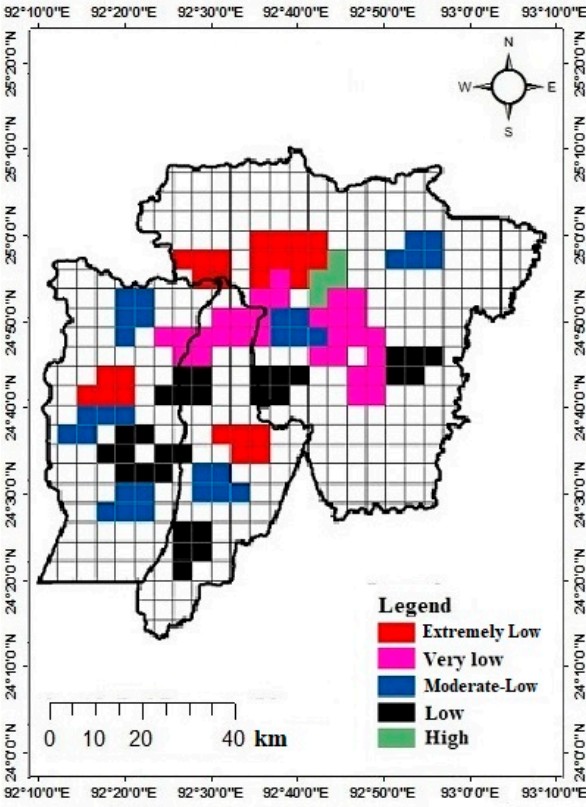

**Figure 13.** The flood resilience map of the housing infrastructure in Barak Valley.

## 5. Conclusions, Limitations, and Future Research Direction

This work illustrates a BBN model for flood resilience quantification of the housing infrastructure system in Barak Valley of Northeast India. The main challenge faced during the quantification of resilience values was a lack of proper post-disaster data. To overcome the challenge, a flood resilience assessment form was developed and a detailed survey was performed in various vulnerable places according to the DDMA, and the resilience values were evaluated for those places. Those quantified values provide a realistic scenario of the housing infrastructure system of this valley, which supports the planners, designers, policymakers, and stakeholders of this valley to become involved in the detailed identification of resilience-influencing parameters for housing systems and the preparedness of these vulnerable places against flood hazards. In this study, influential parameters for resilience quantification were considered and localized and could be mapped with global resilience quantification, which may include a new study area. Lastly, a sensitivity analysis was performed to find the most crucial parent nodes of recovery and reliability, which can also help decision-makers of this valley to focus on the most sensitive parent parameters; to improve a child node, it was not necessary to improve all the associated parent nodes because small changes in the most sensitive parent parameters may lead to a targeted probability of the child node. This analysis will also help in the decision-making process for preparedness against future hazards [86]. As this method is generalized, it can be integrated with any kind of infrastructure system and can be used by a public authority for the resilience quantification of any infrastructure system against flood hazards. The main contributions of the proposed resilience model were: (i) the BBN model provided a more realistic scenario of housing infrastructure for this valley based on collected real data and can be updated by including more uncertain parameters and associated data, and (ii) the sensitivity analysis helped to identify the crucial parent parameters of recovery and reliability against flood hazards.

The following are the recommendations for improving the resilience of the housing/building infrastructure system of this valley against flood hazards based on the discussions with affected householders during the field survey: (i) construction of building infrastructure should follow engineering principles; (ii) people should have a solid understanding of reliability and recovery processes related to housing infrastructure in preparedness for future disasters in the valley; (iii) stakeholders should immediately give attention to the housing infrastructure of Burunga, Poschim Kumrapara, Rajnagar, Panchgram, and Lalaghat, as the resilience of these places was extremely low. Moreover, it was observed that the parent parameters, such as Type_of_house and Wall_thickness were most sensitive regarding reliability, and Insurance and Relief_received were the most vulnerable parent parameters for the recovery of the housing infrastructure against flood hazards. Therefore, decision-makers should strengthen these sensitive parameters to make the infrastructure more resilient against future floods. There are some noted limitations in this work, such as (i) the consideration of more factors for a comprehensive framework is required, (ii) more detailed information about the factors or more data collection is required, and (iii) the involvement of multiple experts from various disciplines is required. In the future, resilience scenarios for other infrastructure systems, such as water, electrical, transportation, and telecommunication systems, as well as critical housing infrastructure systems, such as hospitals, markets, and schools, can be evaluated. Similarly, resilience values against other natural disasters, such as earthquakes and landslides, can be computed [87–89]. Outcome comparisons with other hierarchical-based methods, such as fuzzy AHP (Analytic hierarchy process) and Dempster–Shafer theory, can also be performed. As infrastructure resilience changes with time, these variabilities can be captured with the help of a dynamic Bayesian network.

**Author Contributions:** Conceptualization, M.K.S., S.D. and G.K.; methodology, M.K.S., S.D. and G.K.; software, M.K.S., S.D. and G.K.; validation M.K.S., S.D. and G.K.; formal analysis, M.K.S.; investigation, M.K.S., S.D. and G.K.; data curation, M.K.S.; writing—original draft preparation,

M.K.S.; writing—review and editing, S.D. and G.K.; visualization, M.K.S.; supervision, S.D. and G.K. All authors have read and agreed to the published version of the manuscript.

**Funding:** The first author acknowledges the student's scholarship received from the Ministry of Human Resource and Development, Government of India. The third author acknowledges the financial support through Natural Science Engineering Research Council, Canada Discovery Grant Program (RGPIN-2019-04704) for the professional editing, proofreading, and article processing fees.

**Institutional Review Board Statement:** Not applicable.

**Informed Consent Statement:** Informed consent was obtained from all subjects involved in the study.

**Data Availability Statement:** The sample data collection sheet and the data used for analysis, along with the associated computer programs, can be available on request from the corresponding author.

**Acknowledgments:** The authors acknowledge the support of the District Disaster Management Authority (DDMA), Assam, India, for providing the information of flood-vulnerable places of Barak Valley in Northeast India, as well as their valuable role in providing experts judgment.

**Conflicts of Interest:** The authors declare no conflict of interest.

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
