# Peer review of "Flood Resilience of Housing Infrastructure Modeling and Quantification Using a Bayesian Belief Network"

_sustainability, doi:10.3390/su13031026_

Round 1

Reviewer 1 Report

General comments

The authors adopted four actions as the aim of their work. The first one is to find/identify the factors that most affect the resistance of the housing infrastructure system to flooding. The second aim of the work is to develop a probabilistic model to calculate the resistance to flooding of a housing infrastructure system. The third aim of the work is to calculate the resistance during flooding. The fourth goal of the work is to check the sensitivity of each of the assessed parameters.

The structure of this article is very complex. There are many design assumptions and patterns. However, it is difficult to find a premise in the entire study that would qualify the article for publication in the sustainability journal. I suggest that the authors complete the “sustainability” context in article especially in Introduction section. I have listed a specific remarks below.

Detailed comments

The summary should be improved. Summarize the main findings of the article and indicate the main conclusions and briefly interpret.

Arrange the structure of the article. Define the term "socio-physical infrastructure" (used in the title of subchapter 2).

For the sake of international readers, outline the location of the research area on a national and regional scale. Is there a division into administrative units in Fig. 4 and following?

The figures in the paper are not very clear. Since Figure 4 their layout could be improved by removing empty spaces and enlarging the main content. With what accuracy are the units in the legend of figure 4 (giving 8 decimal places is not correct - what is the part of the meter and what is the significance for the research?). Unify the accuracy for all given values. How was the division of the full scale of values into classes determined? (see legends in figures 4, 7, 8, 9). Consider combining the contents of the figures in order to reduce their number.

Line 163 - Figure 5 - what is shown in the figure: population as in legend or population density as in description? Demographic phenomena are usually presented in the form of a range. According to which principle was the punctual interpretation of this phenomenon adopted in the case of your article? How was this variable introduced into the model?

Lines 270 from 271 - a very obvious statement, it is unreasonable to refer to 6 items of literature in this case [31,66,67,68,69,70].

Before inserting variables (from subsection 3.2) into the model you should check their correlation.

To what end the results are presented in the form of figures 13 and 14. They do not bring new information about the spatial distribution of the studied phenomenon.

This Article should be supplemented by a chapter on test methods, in which the algorithms used will be presented in an orderly manner. This will introduce the transparency of the article, allow you to see the structure of the studies, at what stage the expert method was introduced, at what stage known methods of modeling or calculation of probability were used.

Chapter 7 is entitled to results and discussions, but partial results already appear in previous subsections (sensitivity analysis or model validation). In this chapter there is no analysis of the results, there is no discussion of the values obtained and their comparison with studies of other authors, nor with studies conducted in other fields. There is also no answer to the question of the desirability of using the Bayesian Belief Network, as announced in the title of the article. Figures should also be set in this section.

In section 8, you can find more statements that match the performance discussion and relate directly to constraints and future directions.

Author Response

Many thanks for your inspirations and positive evaluation. The structure of the article is modified in the revised manuscript. We believe that your advice and comments help us to improve the quality of the work. We have tried to modify the manuscript based on your suggestions and comments. Please see each reply with actual modification in the attached file.

Reviewer 2 Report

The paper is overall easy-to-follow and covers a topic that is central in the journal’s scope. A Bayesian Belief Network (BBN) model is developed for resilience quantification of the housing infrastructure against flood hazard. After model validation and a sensitivity analysis that identifies the critical parameters for reliability and recovery of a housing infrastructure system, the proposed framework is applied to the housing system of the Barak Valley community (North-Eastern India). Given the lack of proper post-disaster data, a flood resilience assessment form is developed by the Authors and a detailed survey is carried out in various vulnerable places according to the District Disaster Management Authority (DDMA). Reliability, recovery and resilience against flood hazard are evaluated. It is found that i) especially several places in the valley deserve attention from the stakeholders because they show a very low flood resilience, ii) the reliability is most sensitive to the type of house and wall thickness, and iii) the recovery is most sensitive to insurance and relief received.

Despite the work provides interesting findings, the Reviewer thinks that a major revision is needed. The detailed comments are presented below:

  1. This section is very well organized and the literature review is quite comprehensive. However, some relevant and recent works on resilience assessment are missing and the Authors may want to cite them. Examples of these works are:
    • Marasco, S., Cardoni, A., Noori, A. Z., Kammouh, O., Domaneschi, M., Cimellaro, G. P. (2020). Integrated platform to assess seismic resilience at the community level. Sustainable Cities and Society, 64, 102506.
    • Franchin, P., Cavalieri, F. (2015). Probabilistic assessment of civil infrastructure resilience to earthquakes. ComputerAided Civil and Infrastructure Engineering, 30(7), 583-600.
    • Cavallaro, M., Asprone, D., Latora, V., Manfredi, G., Nicosia, V. (2014). Assessment of urban ecosystem resilience through hybrid social–physical complex networks. ComputerAided Civil and Infrastructure Engineering, 29(8), 608-625.
  2. Section 2. The cost or income values expressed in INR could also be expressed in another currency that is commonly referred to all over the world, e.g. US dollars, for reference.
  3. Section 2. More information, e.g. about the material, could be added about the Assam type houses.
  4. Figure 7. In the legend, “Water point” could be replaced with “Water source”.
  5. Section 3.1. In the current literature, there are additional relevant works (that should be cited) about the use of BBNs in risk and reliability assessment of infrastructure systems. Examples are given by:
    • De Risi, R., Jalayer, F., De Paola, F., Lindley, S. (2018). Delineation of flooding risk hotspots based on digital elevation model, calculated and historical flooding extents: the case of Ouagadougou. Stochastic environmental research and risk assessment, 32(6), 1545-1559.
    • Gehl, P., Cavalieri, F., Franchin, P. (2018). Approximate Bayesian network formulation for the rapid loss assessment of real-world infrastructure systems. Reliability Engineering & System Safety, 177, 80-93.
    • Tien, I., Der Kiureghian, A. (2017). Reliability assessment of critical infrastructure using Bayesian networks. Journal of Infrastructure Systems, 23(4), 04017025.
    • Cavalieri, F., Franchin, P., Gehl, P., D’Ayala, D. (2017). Bayesian networks and infrastructure systems: Computational and methodological challenges. In Gardoni, P. editor: Risk and reliability analysis: theory and applications (pp. 385-415). Springer, Cham.
    • Tien, I., Der Kiureghian, A. (2016). Algorithms for Bayesian network modeling and reliability assessment of infrastructure systems. Reliability Engineering & System Safety, 156, 134-147.
    • Bensi, M., Der Kiureghian, A., Straub, D. (2015). Framework for post-earthquake risk assessment and decision making for infrastructure systems. ASCE-ASME Journal of Risk and Uncertainty in Engineering Systems, Part A: Civil Engineering, 1(1), 04014003.
    • Bensi, M., Der Kiureghian, A., Straub, D. (2013). Efficient Bayesian network modeling of systems. Reliability Engineering & System Safety, 112, 200-213.
    • Bensi, M., Der Kiureghian, A., Straub, D. (2011). Bayesian network modeling of correlated random variables drawn from a Gaussian random field. Structural Safety, 33(6), 317-332.
  6. Lines 325-326. The text says: “flood hazards occur in regular intervals which mainly affect the infrastructure systems and contribute to a considerable rise in annual rainfall in the past few years”. This sentence is not clear. Maybe the Authors mean that a considerable rise in annual rainfall in the past few years contributed to flood hazard. Please check.
  7. Equation 3. A different letter should be used for the state of the varying variable, i.e. p, since it can be confused with “p” meaning probability.
  8. Lines 443-444. It is not clear why the parameters Type_of_house and Wall_thickness are included in the BBN for recovery. This is not consistent with Figure 15(b) (where only six parameters are shown) and the BBN description in Section 3.2.
  9. Line 490. It should be Figures 16, 17 and 18.
  10. A text proofreading is needed to improve readability and to fix several errors/typos (e.g. Line 351: “Validation of a model is (a) critical”; “Resi(s)lience” in the caption of Figures 13 and 14; in the caption of Figure 14, it should be E-2, not E-1).

Author Response

Many thanks for your inspirations and positive evaluation. We believe that your advice and comments help us to improve the quality of the work. We have tried to modify the manuscript based on your suggestions and comments. Please see each reply with actual modification in the attached file.

Round 2

Reviewer 1 Report

The structure of the article is more clear in the current version. The Sustainability thread has been introduced to the articlenew references have been added. The methods used in the paper have been correctly explained, the research area has been describedalthough it is a pity that it is not presented in the figure. 

However, the authors should still improve the abstract. As written in the first review please: 'Summarize the main findings of the article and indicate the main conclusions and briefly interpret.' 

You should improve the quality and aesthetics of the figuresUnify fonts and their sizes. 

Author Response

The structure of the article is more clear in the current version. The Sustainability thread has been introduced to the article, new references have been added. The methods used in the paper have been correctly explained, the research area has been described, although it is a pity that it is not presented in the figure. 

Response: Thank you very much for the positive response.

However, the authors should still improve the abstract. As written in the first review please: 'Summarize the main findings of the article and indicate the main conclusions and briefly interpret.' 

Response: The abstract is modified in the revised manuscript.

You should improve the quality and aesthetics of the figures. Unify fonts and their sizes. 

Response: Thank you very much for the advice. Figures 5, 6, 7, 11, 12 and 13 are uniformly rectified in the revised manuscript.

Reviewer 2 Report

The Reviewer thanks the Authors for addressing all the concerns raised. The manuscript is now improved. However, there are still a couple of minor issues that should be considered before possible publication.

  • In equation 3, it should be p(r|q), as explained in the text, in place of q(r|q).
  • The caption of Figure 10 could read: Sensitivity analysis for (a) reliability and (b) recovery.

Author Response

The Reviewer thanks the Authors for addressing all the concerns raised. The manuscript is now improved. However, there are still a couple of minor issues that should be considered before possible publication.

Response: Thank you very much for the positive response.

  • In equation 3, it should be p(r|q), as explained in the text, in place of q(r|q).

Response: Thank you for the comment. The equation is rectified in the revised manuscript.

  • The caption of Figure 10 could read: Sensitivity analysis for (a) reliability and (b) recovery.

Response: Thank you for the comment. The caption of Figure 10 is rectified in the revised manuscript.